# mtR_find: A Parallel Processing Tool to Identify and Annotate RNAs Derived from the Mitochondrial Genome

**DOI:** 10.3390/ijms24054373

**Published:** 2023-02-22

**Authors:** Asan M. S. H. Mohideen, Steinar D. Johansen, Igor Babiak

**Affiliations:** Genomics Group, Faculty of Biosciences and Aquaculture, Nord University, P.O. Box 1490, 8049 Bodø, Norway

**Keywords:** mitochondria, mitochondrial long non-coding RNAs, mitochondrial small RNAs, mitochondrial tRFs, multiprocessing, read count algorithm, small RNA tool

## Abstract

RNAs originating from mitochondrial genomes are abundant in transcriptomic datasets produced by high-throughput sequencing technologies, primarily in short-read outputs. Specific features of mitochondrial small RNAs (mt-sRNAs), such as non-templated additions, presence of length variants, sequence variants, and other modifications, necessitate the need for the development of an appropriate tool for their effective identification and annotation. We have developed mtR_find, a tool to detect and annotate mitochondrial RNAs, including mt-sRNAs and mitochondria-derived long non-coding RNAs (mt-lncRNA). mtR_find uses a novel method to compute the count of RNA sequences from adapter-trimmed reads. When analyzing the published datasets with mtR_find, we identified mt-sRNAs significantly associated with the health conditions, such as hepatocellular carcinoma and obesity, and we discovered novel mt-sRNAs. Furthermore, we identified mt-lncRNAs in early development in mice. These examples show the immediate impact of miR_find in extracting a novel biological information from the existing sequencing datasets. For benchmarking, the tool has been tested on a simulated dataset and the results were concordant. For accurate annotation of mitochondria-derived RNA, particularly mt-sRNA, we developed an appropriate nomenclature. mtR_find encompasses the mt-ncRNA transcriptomes in unpreceded resolution and simplicity, allowing re-analysis of the existing transcriptomic databases and the use of mt-ncRNAs as diagnostic or prognostic markers in the field of medicine.

## 1. Introduction

Mitochondria are organelles present within all eukaryotic cells, performing oxidative phosphorylation [1] and apoptosis processes [2], among others. Metazoan mitochondria possess their own genomes, which are relatively small (usually 15–20 kb) and contain 14 to about 40 genes, typically 37 in vertebrates [3]. Owing to the multiple cellular copies of mitochondrial DNA, the abundance of mitochondrial transcripts can range from 5 to 30% (depending on the cell type) of the total cellular RNA [4,5]. Mitochondrial non-coding RNAs (mt-ncRNAs) are referred as those encoded in the mitochondrial genome, although nuclear genome-encoded non-coding RNAs (ncRNAs) can be present in mitochondria [6]. Both mitochondrial small non-coding RNA (mt-sRNA) and long non-coding RNA (mt-lncRNA) have been identified both inside mitochondria and in other cellular compartments, and some of their implicated gene regulatory functions have been proposed [4,5,7,8,9]. Despite the growing evidence of regulatory functions of mt-ncRNAs, no appropriate bioinformatic tools to identify them are available up to date.

There are tools such as MITOS [10] or DOGMA [11] to annotate mitochondrial genome, but these tools cannot identify and quantify mt-ncRNAs. Although DOGMA can annotate nucleotide sequences to the mitochondrial genome, the tool requires the entire mitochondrial genome sequence as input and does not work with mt-ncRNAs, which are much shorter. The current analysis of the high-throughput sequencing data relies on the use of tools designed for the nuclear genomic RNA. These tools, as well as DOGMA, cannot identify mt-sRNAs effectively for mt-sRNAs, and frequently have non-templated additions, as well as sequence and length variants [12]. Tools such as tDRmapper [13], SPORTS [14], or MINTmap [15] can be used to analyze mitochondrial tRNA derived fragments (mt-tRFs). However, there is no tool to simultaneously analyze all small RNAs (sRNA) mapping to the mitochondrial genome.

Most tools designed for small RNA data analysis deploy a three-step procedure with some minor modifications [16]. This includes: (1) read count generation, (2) mapping the unique set of sequences to a reference FASTA, and (3) parsing the mapped output files. Read count generation is the most time-consuming step, but it can be significantly reduced by parallelizing the processes on all the available CPU cores. We have developed mtR_find, a bioinformatic tool for identification, annotation and analysis of mtRNA in new or existing transcriptomic datasets produced in any type of sequencing technology. mtR_find uses PYTHON’s multiprocessing functionality that helps to parallelize the analysis of multiple sequencing files for read count generation, thereby massively reducing the data processing time. Along with the tool, we propose a nomenclature to encompass the mt-RNA specificity. The tool allows retrieving the important biological information from the existing datasets in a high-throughput mode in an unpreceded efficiency.

## 2. Results

### 2.1. Performance

The total read counts for the three datasets were: 332.3 million (dataset-1, sRNA-seq of liver samples from malignant tumor tissue of HCC patients and non-malignant tissue from uninfected individuals), 318.2 million (dataset-2, sRNA-seq of semen samples from lean versus obese men), and 93.4 million (dataset-3 (RNA-seq of mouse oocytes); Appendix A). The sRNA datasets were analyzed through parallel processing by mtR_find, and the total execution time for datasets 1 and 2 was 3 min 44 s and 2 min 38 s, respectively. For comparison, the total execution time using MINTmap for dataset-1 and dataset-2 was 48 min 2 s and 34 min 7 s, respectively. The mt-lncRNA analysis was not performed using the parallel processing due to pickling limitations in PYTHON multiprocessing module [17]), and the total execution time was 11 min 29 s. The duration of serial execution of datasets 1 and 2 was 9 min 9 s and 11 min 40 s, respectively. Consequently, the serial execution took ~2.75 times longer than the parallel execution, indicating the efficiency of parallel execution. Besides parallel execution, there are other differences in the way the tool handles mt-sRNAs and mt-lncRNAs. The tool does not consider sequences longer than 50 nt for mt-sRNA computation and shorter than 50 nt for mt-lncRNA. For mt-sRNA, every single sequence is considered unique by the tool. For mt-lncRNA, the tool outputs the unique sequence count and, in addition, the counts of lncRNA sequences with same 5′ end but variable 3′ end are summed together. In addition to mt-lncRNAs that are longer than 200 nt, mt-lncRNA option of mtR_find also identifies ncRNAs that are 50–200 nt long, which are categorized as mid-size or intermediate RNAs. In order to study only lncRNAs that are longer than 200 nt, users can use the “—filter 200” argument as a command line option while running mtR_find.

### 2.2. Read Statistics

Datasets-1, -2, and -3 had, respectively, 36,136, 93,128, and 9222 unique sequences with a total read count greater than 200 (Appendix A). The numbers of sequences that mapped to the mitochondrial genome were 2120 (constituting 1.2% of total reads), 8899 (4.4%), and 178 (1.4%), respectively (Appendix A). Out of these, reads mapping to heavy strand composed 71.5%, 67.4%, and 43.5% of the total mitochondria-derived sequences respectively, while the remaining reads mapped to the light strand (Appendix A).

### 2.3. Length Distribution and Annotation of mt-ncRNAs

We found a diverse size range (Appendix A) and gene origins (Appendix A) of mitochondrial non-coding RNAs in the datasets examined. Datasets-1 and -2 were enriched in mt-sRNAs in the size range of 31–32 nt and 27 nt, respectively, while the mt-lncRNAs in the dataset-3 were in the size range of 87 to 141 nt. Most of mt-lncRNAs in the dataset-3 had length variants (Appendix A). The majority of them belonged to three genes, namely, ATP6, ATP8, and CytB (Appendix A).

### 2.4. Differential Expression of mt-ncRNAs

There were differences in number of reads mapping to mitochondrial genes between the subject and control groups in both the dataset-1 and dataset-2 (Appendix A). PCA for mt-sRNAs (Appendix A) and the heatmap of top 50 highly variable read sequences (Figure 1) showed clustering of two different groups consistent with the subject and controls, although there was a small variability within groups resulting from biological replicates.

Differential expression (DE) analysis of mt-ncRNAs was performed on the data from dataset-1 (chronic hepatitis C-associated cancer vs. non-cancer liver samples; chronic hepatitis B-associated cancer vs. non-cancer liver samples; chronic hepatitis C-associated cancer vs. uninfected cancer liver tissue samples; and chronic hepatitis B-associated cancer vs. uninfected cancer liver tissue samples, Table 1) and dataset-2 (semen from obese vs. lean subjects). In the dataset-1, there was a significant reduction (*p* < 0.005) in the relative abundance of tRNA half (tRH) mapping to tRNA genes of nuclear genome origin, namely, tRFs from tRNA^Gly^ and tRNA^Val^ in cancer tissue when compared to non-cancer liver tissue [18]. We observed a similar trend for DE mitochondrial tRHs. For example, when looking to chronic hepatitis C-associated cancer vs. non-cancer liver tissue samples comparison, 13 out of 354 DE tRFs were tRHs and 10 of them were significantly downregulated in the cancer cells (Appendix A). Five of these ten mitochondrial tRHs originated from tRNA^Val^. In the dataset-2, 75 DE mt-sRNAs (39 up- and 36 down-regulated in semen samples from obese vs. lean individuals) were identified, all of them originating from the mitochondrial large subunit rRNA (Appendix A). The majority of them existed as length variants and all of them clustered at a region with sequence start site between 2690 and 2706 in the mitochondrial large subunit (mtLSU) rRNA gene, with 2704 and 2705 being the two most common sequence start sites.

### 2.5. Novel Mitochondrial tRFs and Non-Coding RNAs Detected by mtR_find

The DE mt-tRFs (783 unique mt-tRFs) from the dataset-1 were compared with tRFs downloaded from MINTbase, an extensive database of 28,824 nuclear and mitochondrial tRFs obtained from 12,023 cancer datasets using MINTmap tool [19]. There were 365 (46.6%) tRFs not found in MINTbase, including 214 tRFs-5, 42 tRFs-3, 43 i-tRFs-3, 56 i-tRFs-5, 8 tRNA-half-5, and 2 tRNA-half-3 (Appendix A). All these novel tRFs had normalized reads per million (RPM) value greater than one (Appendix A), a cut-off value in MINTbase.

### 2.6. Performance of the Tool with Simulated Data Set

There were 16 simulated sequences of mt-lncRNA, including 7 from the heavy strand, 5 from the light strand, and 4 antisense to heavy strand genes with substitutions and grouped as light strand transcripts. The simulation gave results concordant with the mtR_find (Appendix A). The CSV files from both the simulation and mtR_find analyses were loaded as data frames using PYTHON pandas module, element-wise comparison was performed between the two data frames, and the results were similar (Appendix A).

## 3. Discussion

mtR_find is the first small RNA tool to incorporate parallel processing by reading multiple input files simultaneously and processing them at the same time. The mtR_find tool performs much better when compared to published small RNA tools such as MINTmap [15]. Results from testing mtR_find on the simulated dataset shows that the sensitivity of mtR_find is high. The read count algorithm of mtR_find can be used for developing tools for the analysis of other sRNA types by replacing the reference and modifying the annotation criteria. Even though the parallel processing significantly reduces the execution time, it has to be noted that the execution time is CPU-dependent. Furthermore, if the number of CPU is not commensurate with the available RAM, the script might run into memory errors. In such a case, a user has to lower the CPU count manually by using the command line parameters to circumvent the issue. The execution time of mtR_find is much lower than MINTmap and also includes the time to download both the GTF file and the mitochondrial genome. If these files are provided manually as input files, then the execution time will be further reduced. Moreover, mtR_find identified 365 tRFs that are not present in MINTbase v2.0. Due to the presence of overlapping reading frames in several mitochondrial genes, mt-sRNA sequence start and end sites of ±3 were used for annotating the mt-sRNAs in our tool; indeed, 266 out of the 365 sequences had sequence start site or end site at ±3 nt from the gene start or end boundary, respectively (Appendix A). And, 42 out of these 266 mt-sRNAs, had sequence start or end site either before or after the 5′ and 3′ end of tRNA gene boundary, respectively. Hence, mtR_find is highly sensitive in capturing all mtsRNAs from the mitochondrial genome.

mtR_find identified features in the test datasets that had not been identified before. mtR_find identified reads mapping to the light strand in the range of 28.5–56.5%. This result is discrepant with the previous studies on mt-sRNAs, where it has been shown that the number of reads from the light strand constituted approximately 3–5% of all the mitochondrial reads [4,12]. Notably, we found a considerable number of reads mapping to the light strand in an anti-sense orientation to the heavy strand genes. Small RNAs derived from a nuclear genome are classified based on their biogenesis pathways, and the length of small RNAs acts as a proxy indicator for biogenesis. For example, tRNA half (tRH), miRNAs, and piRNAs are typically 32–34 nt, 21–22 nt, and 26–31 nt in length, respectively, in most studied species [20]. A quick review of the findings from the original studies (dastasets-1 and -2; [18,21]) revealed that these datasets were enriched in tRHs and piRNAs of nuclear genome origin, respectively. Interestingly, we found that a majority of mt-sRNAs in the dataset-1 were tRH of 31–32 nt length, and this frequency of mitochondrial tRH was strikingly similar to that of nuclear tRH [18], suggesting a similar biogenesis pathway. In the case of dataset-2, majority of mt-sRNAs of 27 nt size mapped to mt-rRNA. Although the size range is indicative of piRNA biogenesis, there is only a single study showing the localization of PIWI proteins as well as piRNAs mapping uniquely to the mitochondrial genome [22]. We found the sequence start sites of these putative 29 mitochondrial piRNAs [22] either exactly overlapped or were in the proximity of ±3 nt of sequence start sites of 27-nt mt-sRNAs from the dataset-2. However, it is not known whether these mt-sRNAs are processed through a particular biogenesis pathway with a defined biological function. Except for tRFs, no curated database exists for mitochondria-derived sRNAs or ncRNAs. Therefore, all the remaining differentially expressed mt-RNAs from datasets 1 and 2, have been not catalogued before. In case of mt-lncRNAs in dataset-3, the majority of sRNAs were derived from ATP6, ATP8, and CytB. lncCytB is among the most abundant mitochondrial lncRNAs in HeLa cells [23] and its abnormal trafficking has been demonstrated in human hepatocellular carcinoma cells [24]. To our knowledge, other mt-lncRNAs found in mouse oocytes and 1-cell embryos (dataset-3) have no functional annotations yet.

mtsRNAs identified in datset-1 and daaset-2 might have biological implications. The abundance of tRH of nuclear genome origin is positively correlated (Spearman’s rho = 0.67–0.87) with angiogenin mRNA/protein abundance in non-cancer liver tissue [18]. Differences in the expression of nuclear genome-derived tRFs produced through enzymatic cleavage of angiogenin have been observed [25]. These nuclear genome-derived tRFs bind to cytochrome C (a protein complex partially encoded by the mitochondrial genome) to prevent cells from undergoing apoptosis [25] and it has also been showed that these tRFs improve cell survival by acting in response to stress [26,27]. Although it is unknown whether tRFs of mitochondrial origin act in a similar way, differences in the expression of mitochondrial non-coding RNAs have been associated with cancer [8,28,29]. Moreover, it has been shown that the processing of the mitochondrial tRNAs at both the 5′ and 3′ ends has a substantial effect on mitochondrial gene expression [30,31]. Since mitochondrial tRFs are generated from both the 5′ and 3′ end of the mitochondrial tRNAs, and aberrant expression of mitochondrial genes leads to many disease conditions including cancer, DE mitochondrial tRFs in dataset-1 could potentially be implicated to disease condition. In dataset-2, the authors have indicated that differences in expression of piRNAs between spermatozoa from lean and obese men may increase the chances of offspring to develop obesity. No studies investigating the expression of mt-sRNAs in obesity are available; however, it has been shown that mitochondrial peptides are involved in regulating metabolism [32]. The expression of mitochondrial peptides is hypothesized to be controlled by mt-sRNAs [4]. Hence, altered expression of mt-sRNAs may result in an impaired metabolic pathway, which, in turn, might result in obesity.

Interestingly, no single mt-sRNA mapped to the termination association sequence (TAS) in the mitochondrial DNA control region, neither in the dataset-1 nor in the dataset-2. Small RNAs originating from the TAS region (co-ordinates 16,161 to 16,188 in the mouse mtDNA sequence) within the mitochondrial control region were expressed in mice [33].

Studies on tRFs have shown that a disproportionately high number of unique tRFs was derived from mitochondrial tRNA genes (*n* = 22) when compared to nuclear tRNA genes (*n* = 625) in humans [34,35]. For example, a study on samples from prostate cancer patients demonstrated that 62.0% tRFs originated from nuclear tRNA genes, while the remaining 38% originated from the mitochondrial tRNA genes [35]. This indicates the diversity of mitochondrial tRFs. Many of these mt-sRNAs map uniquely to the mitochondrial genome and not to the mitochondrial DNA-like sequences (NUMTs) in the nuclear genome [36]. Moreover, it has been shown that expression of mt-sRNAs is not associated with levels of NUMT but varies across different tissues depending on the mitochondrial DNA content [36]. This indicates mt-sRNAs have biological roles and, hence, mt-sRNAs were found to be differentially expressed in dataset-1 and 2 could be implicated in disease condition.

## 4. Materials and Methods

### 4.1. Implementation

The code for mtR_find is written in PYTHON 3.6.8 (also compatible with PYTHON 2.7.5) and requires dependencies that include PYTHON modules: pandas (version 0.21.0 and above) [37], multiprocessing, matplotlib [38] (optional) and other tools such as bowtie (version 1.1.2 and above) [39] and samtools (version 1.9 and above) [40].

### 4.2. Data Resources, Extraction of Mitochondrial Genome, and Annotation File

Depending on the species of interest (input parameter), mitochondrial genomes of *Homo sapiens*, *Danio rerio*, *Gallus gallus*, *Mus musculus*, and *Rattus norvegicus* have been downloaded from Ensembl [41]. In the case of *Xenopus laevis* and *Xenopus tropicalis*, the mitochondrial genomes have been downloaded from Xenbase [42]. A bowtie index corresponding to the particular genome was created using default parameters. The gene annotations were obtained by downloading the gene transfer format (GTF) annotation file for the species of interest from Ensembl/Xenbase and extracting the information pertinent to the mitochondrial genes. For any other species not listed above, the FASTA and GTF files have to be downloaded and provided manually by the user. The script mt_annotaion.py is useful to pre-process the GTF file (https://github.com/asan-nasa/mtR_find/blob/master/add-on/mt_annotation.py, accessed on 26 August 2022).

### 4.3. ncRNA Count Generation

In the ncRNA-count generation step, a dictionary of unique sequences was created from the list of all input FASTQ files. Using this as a reference, the count number for each unique sequence was determined for individual FASTQ files. The default cut-off threshold value for sequences is <200, because the counting accuracy of low ncRNA-count sequences can be erratic [5,43]. However, users can specify their own cut-off value tailored for the specific needs of their analyses. The output read count file is in comma separated value (CSV) format, in which the row names are unique sequences and column names are file names. Individual rows display the count number of a particular sequence in the corresponding library. In the case of SOLiD sequencing data, reads have to be mapped to the corresponding genome and converted from color-space to FASTQ files using adapt_find script [44], available at https://github.com/asan-nasa/adapt_find/blob/master/adapt_find.py (accessed on 26 August 2022) prior to the read-count generation step.

### 4.4. Mapping

Unique sequences from the read count file were extracted, converted to FASTA format, and mapped against the mitochondrial bowtie index using the following parameters: bowtie --best –v 1 –p 20. The mapped and unmapped sequences from the resulting SAM file were filtered out using samtools. Unmapped sequences carrying a non-templated CCA motif at their 3′ ends were retrieved, the CCA motif was trimmed, and the sequences were again mapped to the mitochondrial genome, this time under zero-mismatch stringent criterion to avoid false positive findings. The sequences mapping to the 3′ end of mitochondrial tRNA genes in the sense direction or to the 5′ end in the anti-sense direction were annotated as having a non-templated CCA additions at their 3′ ends (Figure 2).

### 4.5. Annotation

Genomic locations of mapped sequences were determined (Figure 3). Then, the gene annotation was performed using individual mitochondrial genes (Appendix A). The final sequence annotation was based on the position of a mapped sequence and its length within a gene using the MINTbase criteria [19] with some modifications (Appendix A). For both mt-sRNA and mt-lncRNA, if the sequence start site is in one gene and the end site is in another gene (Figure 3D), the gene that has the sequence start site is taken for annotation. The only exception to this rule is tRF-1. MINTbase classification of mt-sRNAs includes tRH-5′ and tRH-3′, and tRNA derived fragments (tRFs) include tRF-5′, tRF-3′, tRF-1, and i-tRF.

### 4.6. Nomenclature

Two levels of ID were produced. The specificID provides a unique annotation for every possible isoform of a sequence. The general ID provides the annotation of the family the given sequence belongs, in the terms of typical starting nucleotide, and skipping information on the sequence length and modifications from the main form. The nomenclature format for mt-sRNA is: “species_name”|”mt-sRNA”|”gene”|”sequence subtype”|”Strand”|”Orientation”|”Sequence start position”|”Sequence length”|Substitutions. For mt-lncRNA, the format is “species name”|”mt-lncRNA”|“gene”|”strand”|”sequence start position”|”sequence length”.

The species abbreviation is a three- or four-letter organism code as proposed in Kyoto Encyclopedia of Genes and Genomes (www.genome.jp/kegg/catalog/org_list.html (accessed on 19 February 2023)). The species abbreviations used in the present study are given in Appendix A. Gene name refers to one of the mitochondrial genes (Appendix A). If the sequence falls in a non-coding region, then it is denoted as “non-coding (“nc”) (Figure 3). The sequence subtype refers to the specific location in a gene transcript (applicable only for mt-sRNAs), as defined in Appendix A. Sequence start position refers to the genomic position of the 5′ nucleotide of the sequence. Strand refers to either heavy or light strand. Antisense orientation indicates anti-sense mapping of the sequence to a particular gene. Substitutions refer to any mismatches in the sequence as compared to the reference genome; if they occur, nucleotide position (from the start of the sequence) is given, along with the base letter to which the main form has been altered. The example nomenclature is given in Table 2.

### 4.7. Training-Experimental Dataset

We tested the tool on two small RNA (sRNA) datasets [18,21] downloaded from NCBI, and one long non-coding RNA dataset (unpublished study [45]) downloaded from European Nucleotide Archive (ENA). MINTmap was also tested on the two sRNA datasets to compare the performance of mtR_find with that of MINTmap. The two sRNA datasets were generated in studies where mt-ncRNAs were not analyzed. The dataset-1 contained information from sRNA-seq of hepatocellular carcinoma (HCC) versus non-malignant liver samples from subjects with chronic hepatitis B or C (*n* = 4 for each group), as well as uninfected subjects undergoing resection of metastatic tumors control group (*n* = 4, Appendix A). In the dataset-2, the information was obtained from sRNA-seq of semen samples from 23 human subjects, classified as either lean (*n* = 13) or obese (*n* = 10; Appendix A). The dataset-3 has been generated from RNA-seq of mouse oocytes (*n* = 2) and 1-cell embryos (Appendix A).

In the case of sRNA datasets, the SRA files were downloaded using prefetch SRA utility tool. The SRA file format was converted to FASTQ files using fastq-dump tool [46]. Adapter sequences were removed from the raw FASTQ files, bases with quality score less than 20 were trimmed from the 3′ end. Sequences shorter than 15 nt were removed. The read count of mt-sRNA sequences was extracted by running mtR_find and differential expression analysis was performed using DESeq2 R package [47]. mt-sRNA sequences with a Benjamini–Hochberg adjusted *p*-value of <0.1 were considered differentially expressed (subject versus control). For mt-lncRNAs, paired-end FASTQ files obtained from ENA were converted to single-read FASTQ files using FLASH [48] and then run on the mtR_find tool. Due to the lack of biological replicates in the dataset-3, only the relative abundance of read counts was reported in our analysis.

### 4.8. Training-Simulated Dataset

mtR_find was tested on simulated datasets for both mt-sRNA and mt-lncRNA using separate scripts with the following command line parameters: (1) FASTA file (in this case, zebrafish mitochondrial genome); (2) GTF file (zebrafish mitochondrial gene annotation information); (3) desired number of unique sequences in each stimulated file; and (4) total number of stimulated files to be created. The GTF file was read and separated into two lists. The first list was based on the strand specificity: heavy strand or light strand, while the second one was based on genes.

The simulation script picked a random sequence start position from a random gene or from the non-coding region, in either the heavy or the light strands. Then, a random length was selected and added to the sequence start position to compute the sequence end position. Using the sequence start- and end-positions as co-ordinates, the sequence was extracted from the input mitochondrial genome. For the light strand sequences, the reverse compliment of the forward strand sequence was extracted, and a random count number for this particular sequence was assigned for each simulated file. This information was then used to create a simulated FASTQ file using the sequence and count information for each sequence. Random simulation of sequences and the corresponding read counts was performed using PYTHON module “random”. The simulation script outputs a simulated read count CSV file with sequence and annotation information, which should match the output of the mtR_find when the simulated FASTQ files are being analyzed.

Simulation scripts used different strategies to distribute reads among different sequences as described in Appendix A. However, in both methods the total number of reads was split in such a way that 80–95% were simulated from the heavy strand and the remaining 5–20% were from the light strand. The simulated dataset has been tested using mtR_find tool, and the results were compared with the results from the simulation. The four different parameters were calculated to check the concordance: (1) number of unique sequences; (2) sequences mapping to the mitochondrial genome and the distribution of sequences between the two strands; (3) total read count and count of individual sequences in each file; and (4) annotation information and read count distribution among four bio-types. The bio-types included rRNA, tRNA, non–coding region, and protein-coding genes.

Simulation and testing of the tool were performed on a Linux server (Red Hat 4.8.5–28) with Python 3.6.8 (64 CPU cores, 504 GB RAM).

### 4.9. Identification of Novel tRFs

tRFs were downloaded from MINTbase [19] as a tab delimited file, while the mitochondrial tRFs (test sequences), obtained from mtR_find, were in CSV format. Both files were loaded as separate pandas data frames and the sequence column was extracted into two separate lists. Then, the sequences from the two lists were compared (Appendix A). Only exact sequence matches were allowed.

## 5. Conclusions

Existing tools can identify only a sub-group of mtsRNAs. mtR_find is the first publicly available tool to comprehensively analyze and annotate all mitochondrial non-coding RNAs. The novel read count algorithm significantly reduces the execution time, making a high-throughput analysis of multiple datasets possible. mtR_find does not create any intermediate files and, hence, saves disk space. Moreover, mtR_find generates a single script for pre-processing data, mapping reads, and then generating count data with annotation information for files. mtR_find identifies novel mt-sRNAs, such as tRFs or mt-lncRNAs, in the existing datasets. It opens a new analytical possibility to re-examine thematic RNA-seq clusters of datasets in search for novel diagnostics markers.

## Figures and Tables

**Figure 1 ijms-24-04373-f001:**
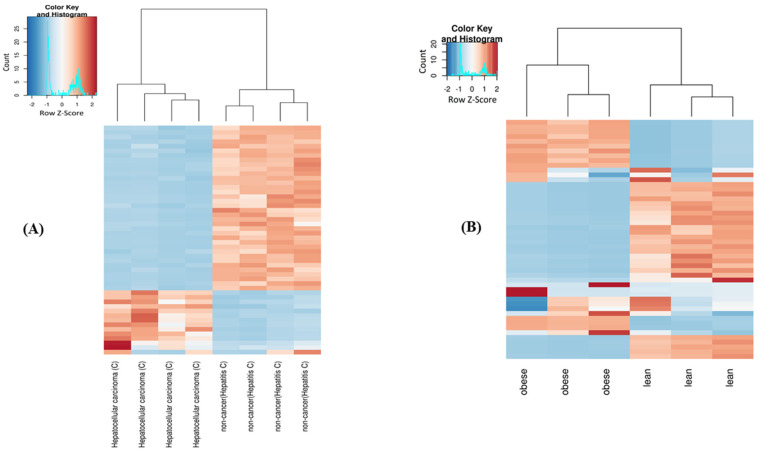
Clustering of the top-50 highly variable small mt-ncRNAs in the dataset-1 (**A**) and dataset-2 (**B**).

**Figure 2 ijms-24-04373-f002:**
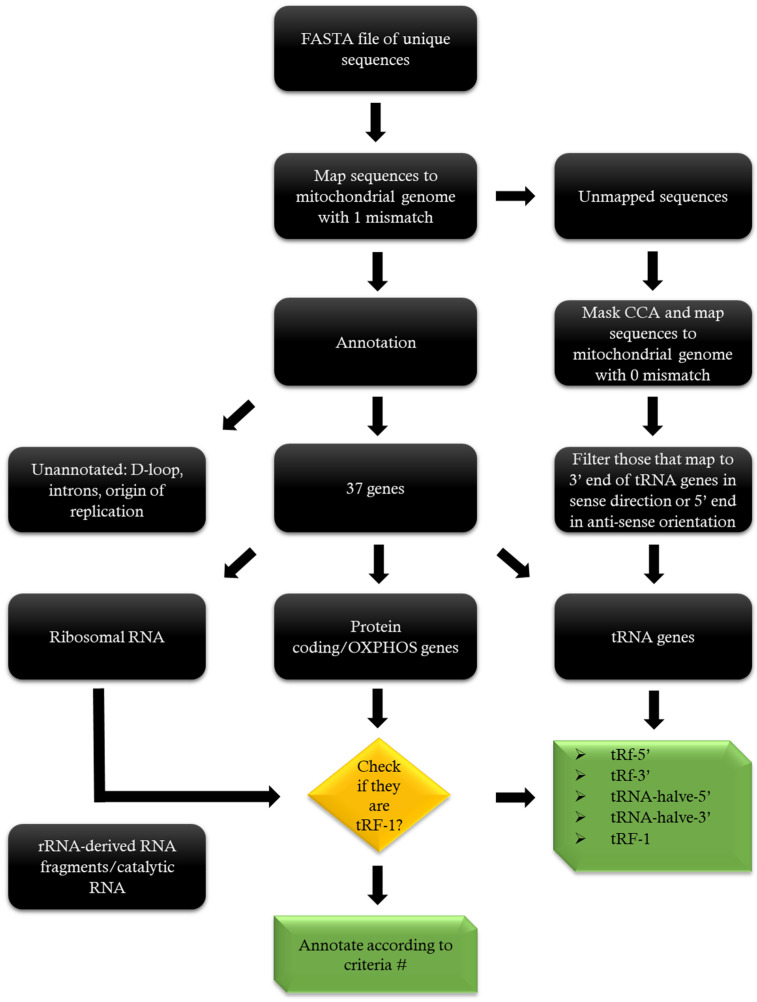
Workflow of mapping and annotation of mt-sRNA. # Annotation criteria: please see Appendix A.

**Figure 3 ijms-24-04373-f003:**
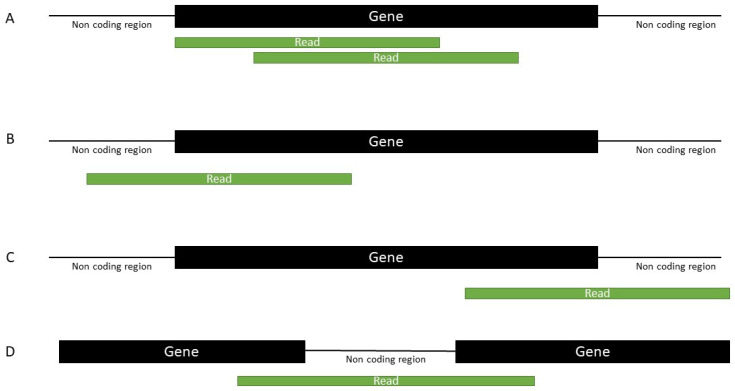
Possible alignments of a mt-ncRNA: (**A**), falls within a gene boundary; (**B**), starts in a non-coding region and overlaps a gene boundary; (**C**), falls in a non-coding region. (**D**), overlaps two gene boundaries.

**Table 1 ijms-24-04373-t001:** Differentially expressed (DE) mt-sRNAs (*p*-adj < 0.1) in pairwise comparisons of datasets generated by [18] (dataset-1). Liver samples (*n* = 4 for each group) were obtained from subjects suffering chronic hepatitis C (HC) or hepatitis B (HB), with either diagnosed hepatocellular carcinoma or non-malignant tissue, as well as from uninfected subjects with hepatocellular carcinoma. Arrows show significantly (*p*-adj < 0.1) up-or down-regulated mt-sRNAs.

Comparison	Total DE			tRNA			rRNA			Non-Coding			Protein-Coding			Log2foldchange	
	Total	↑	↓	Total	↑	↓	Total	↑	↓	Total	↑	↓	Total	↑	↓	Min	Max
HC cancer vs. uninfected	423	224	199	348	216	132	53	6	47	11	2	9	11	0	11	−25.88	7.05
HB cancer vs. uninfected	369	206	163	304	154	150	55	48	7	4	0	4	6	4	2	−25.1	8.6
HB cancer vs. non-cancer	369	208	161	265	143	122	82	51	31	9	6	3	13	8	5	−8.15	7.5
HC cancer vs. non-cancer	437	255	182	354	220	134	56	22	34	13	11	2	14	2	12	−12.12	10.15

**Table 2 ijms-24-04373-t002:** Examples of nomenclature for mt-sRNA and mt-lncRNA with individual field separator values. The column “Specific-ID” shows the nomenclature and the values inside the field separators correspond, in order, to the values in the column names “Species” through “Substitutions” (from left to right).

	Species	ncRNA	Gene	Sequence Subtype	Strand (H or L)	Orientation (Sense or Anti-Sense)	Sequence Start Position	Sequence Length	Substitutions	Specific-ID
mtsRNA	hsa	mt-sRNA	Glu	tRH-3	L	sense	14,676	34	NIL	hsa|mt-sRNA|Glu|tRH-3|L|14676|34
	dre	mt-sRNA	Glu	tRH-3	L	anti-sense	14,675	32	NIL	dre|mt-sRNA|Glu|tRH-3|L|as14675|32
	mmu	mt-sRNA	Arg	tRF-5	H		10,406	25	24C0	mmu|mt-sRNA|Arg|tRF-5|H|10406|25
mtlncRNA	rno	mt-lncRNA	ND1		L		3310	201		rno|mt-lncRNA|ND1|L|3310|201
	hsa	mt-lncRNA	COI		H		6015	150		hsa|mt-lncRNA|COI|H|6015|150
	xen	mt-lncRNA	ATP6		L		8550	85		Xen|mt-lncRNA|ATP6|L|8550|85

## Data Availability

Datasets used in the study are downloaded from NCBI and information is included in Appendix A. The script for mtR_find is available at https://github.com/asan-nasa/mtR_find/blob/master/mtR_find.py (accessed on 26 August 2022).

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
