# Peer review of "mtR_find: A Parallel Processing Tool to Identify and Annotate RNAs Derived from the Mitochondrial Genome"

_ijms, 2023, doi:10.3390/ijms24054373_

Round 1

Reviewer 1 Report

I wasn't able to use the tool when trying to test it. Both the documentation in the paper and at https://github.com/asan-nasa/mtR_find fail to correctly state the required parameters to run the tool. The code to check the pandas version is incorrect, declaring that pandas v 1.2.3 was < 0.21.0. The lack of an option to suspend multithreading means it cannot be run in docker containers that are restricted from launching additional threads. It is at present not in fit state for public release to the research community.

The flow of the paper would be improved if on line 67, when describing the size of the test datasets, the authors identified what those dataset are. The difference in the tool’s handling of sRNA and lncRNA datasets should be described.

Figures S4 and S5 are unlabelled. Line 94 refers to figure S5A when it should be S5C.

On line 125, “1,25,285” is a typo.

The list of dependencies on line 246-247 do not match those on the github readme.

The use of the phrase “read count” in section 4.3 may be confusing – this is typically used to refer to the number of aligned reads overlapping an annotated feature rather than the number of repeats of a particular sequence in an unaligned reads file.

In section 4.5, it appears that Fig. 3 doesn't contribute to how genes are annotated, rather the criteria in Table S3 are used. Therefore, I question the purpose of Fig. 3.

In section 4.6, there are multiple issues with your nomenclature scheme. First of all, you use dashes both within fields and as field separator, which is poor practice.  Secondly, you have placed mt-sRNA and mt-lnc within braces in your definitions (e.g. ) indicating these fields are variables like the others, when in fact they are the verbatim strings that appear in that position – remove the braces. Third, you indicate one field represents “sequence subtype as given in Additional file 8, Table S2”, however S2 is a gene table, not a subtype table. Furthermore, your example IDs don’t appear to follow your defined schema. A table with labeled columns and example values would be more clear.

Author Response

Thank you very much for your comments, they are very helpful.  We revised the manuscript and the code accordingly. Please find below responses to your questions and suggestions:

  • I wasn't able to use the tool when trying to test it. Both the documentation in the paper and at https://github.com/asan-nasa/mtR_find fail to correctly state the required parameters to run the tool. The code to check the pandas version is incorrect, declaring that pandas v 1.2.3 was < 0.21.0.

RESPONSE: We are very sorry for it. The issue with pandas version in the script has been fixed and an updated version of the script is uploaded again in GitHub. The parameters to run mtR_find were previously included in the manual folder of the GitHub page, and in the  help section of the tool. Currently, all the necessary information are included in the “readme” section with required/optional parameters and usage examples (https://github.com/asan-nasa/mtR_find/blob/master/README.md). The previous version of the script might not be running because it exited when it performed version check of Pandas (as found by Reviewer 1). The updated script now runs with pandas v 1.2.3 and it is working fine. For your reference, please see the attached video file (files for Reviewers).

  • The list of dependencies on line 246-247 do not match those on the github readme.

RESPONSE: The updated list of dependencies are added to the “readme” section. Now the dependencies in the paper (Section 4.1., line number 268 to 271) match with that of the dependencies listed in github.

  • The lack of an option to suspend multithreading means it cannot be run in docker containers that are restricted from launching additional threads. It is at present not in fit state for public release to the research community.

RESPONSE: It is fixed now. There is an optional argument “—suspend”. If users want to suspend multiprocessing they can invoke the optional argument “—suspend yes”. The default value is “no”.

  • The flow of the paper would be improved if on line 67, when describing the size of the test datasets, the authors identified what those dataset are. The difference in the tool’s handling of sRNA and lncRNA datasets should be described.

RESPONSE: Necessary changes are made - line number 71-74 and 83-93.

  • Figures S4 and S5 are unlabelled. Line 94 refers to figure S5A when it should be S5C.

RESPONSE: Figure S4 and S5 are now labeled. The reference is corrected from Figure S5A to S5C (line number 112).

  • On line 125, “1,25,285” is a typo.

RESPONSE: Replaced with correct number (line number 143).

  • The use of the phrase “read count” in section 4.3 may be confusing – this is typically used to refer to the number of aligned reads overlapping an annotated feature rather than the number of repeats of a particular sequence in an unaligned reads file.

RESPONSE: The word read count is replaced as “ncRNA-count” (line number 286-287,290)

  • In section 4.5, it appears that Fig. 3 doesn't contribute to how genes are annotated, rather the criteria in Table S3 are used. Therefore, I question the purpose of Fig. 3.

RESPONSE: Table S3 is only for mtsRNA, but not for mtlncRNA. Whereas, Fig. 3 is intended for both mt-sRNA and mt-lnRNA. Fig. 3 depicts the first level of annotation, on which gene to select for annotation when the sequence start and end site falls on different genes (added sentences in line 316-317). Modified version of Fig. 3 is included and appropriate changes are made to Fig. 3 legend.

  • In section 4.6, there are multiple issues with your nomenclature scheme. First of all, you use dashes both within fields and as field separator, which is poor practice.  Secondly, you have placed mt-sRNA and mt-lnc within braces in your definitions (e.g. ) indicating these fields are variables like the others, when in fact they are the verbatim strings that appear in that position – remove the braces. Third, you indicate one field represents “sequence subtype as given in Additional file 8, Table S2”, however S2 is a gene table, not a subtype table.

RESPONSE: Instead of dash, vertical bar (“|”, upright slash) is now used as a field separator. The braces are also removed in the nomenclature section. The reference be Additional file 8, Table S3. It is corrected in the manuscript (line 357).

  • Furthermore, your example IDs don’t appear to follow your defined schema. A table with labeled columns and example values would be more clear.

RESPONSE: We added Table 2 showing example nomenclature.

Reviewer 2 Report

In this study, the authors present a python package suitable for the expression mapping and annotation of non-coding (and coding) RNAs from short RNA-seq reads derived from the mitochondrial genome.

While the topic is interesting, there are several points in the manuscript that remain unclear and need to be resolved.

The computer package maps short reads against the mitochondrial genome. There are also nuclear encoded RNAs in the mitochondria. Therefore, the title is not specific. Another title such as: mtR_find: A parallel processing tool to identify and annotate RNAs derived from the mitochondrial genome.

Line 15: mt_find uses a novel read count algorithm: This is unspecific. Please rephrase for improved clarity.

Line 41: authors must acknowledge that RNAs are also imported and exported from the mitochondria, wherefore mt-sRNA and mt-lncRNA would not necessarily be restricted to RNAs originating from the mitochondrial genome. Hence, the definition should be rephrased and refined.

Line 61: When putting Results before Methods, then it is essential for legibility to include a short description of the principles of the analysis.  Moreover, it does not make sense to separate the Discussion from the Conclusion by inserting Materials and Methods.... I suggest that the authors restructure the manuscript to put materials and methods before results. Although this does not follow the basic template from IJMS, it makes more sense for this paper.

Line 110: define tRH the first time it occurs

Table 1: This table would benefit from more details. What data set are the data derived from? Are all of these significantly regulated? Average degree of regulation?

Line 167: The feature of overlapping reading frames also constitutes a limitation of the software, because the short reads from the next generation sequencing does not give information about the start and end of the non-coding/alternative RNAs.

Line 186: The software computes alignment to the mtDNA. How many of the hits are also present in nuclear DNA? The short read length makes is likely that the sequences are not unique to the mitochondria. Hence, I believe that the authors must quantitify how many of the short RNAs that could also derive from the nuclear genome.

Line 317: lncRNAs are normally defined as being above 200nt and having no coding potential. From Additional File 8 it seems like lncRNAs are defined somewhat different (>87nt)

Author Response

Thank you very much for your comments, they are very helpful.  We revised the manuscript and the code accordingly. Please find below responses to your questions and suggestions:

  • While the topic is interesting, there are several points in the manuscript that remain unclear and need to be resolved. The computer package maps short reads against the mitochondrial genome. There are also nuclear encoded RNAs in the mitochondria. Therefore, the title is not specific. Another title such as: mtR_find: A parallel processing tool to identify and annotate RNAs derived from the mitochondrial genome.

RESPONSE: The title is modified as suggested

  • Line 15: mt_find uses a novel read count algorithm: This is unspecific. Please rephrase for improved clarity.

REPSONSE: Rephrased (lines 15 to 16)

  • Line 41: authors must acknowledge that RNAs are also imported and exported from the mitochondria, wherefore mt-sRNA and mt-lncRNA would not necessarily be restricted to RNAs originating from the mitochondrial genome. Hence, the definition should be rephrased and refined.

RESPONSE: Thank you. Definition is rephrased (lines number 35-41).

  • Line 61: When putting Results before Methods, then it is essential for legibility to include a short description of the principles of the analysis.  Moreover, it does not make sense to separate the Discussion from the Conclusion by inserting Materials and Methods.... I suggest that the authors restructure the manuscript to put materials and methods before results. Although this does not follow the basic template from IJMS, it makes more sense for this paper.

RESPONSE: We do agree to move the Materials and Method above the Result section. However, for the purpose of this revision, when we try to move “results” section above “methods” section, the track changes for the modifications in the manuscript made in this revision won’t be visible for the whole sections. Therefore, we did not change the section order, but we would like to do it at a later stage when all the revisions are complete.

  • Line 110: define tRH the first time it occurs

RESPONSE: Defined in line number 128

  • Table 1: This table would benefit from more details. What data set are the data derived from? Are all of these significantly regulated? Average degree of regulation?

RESPONSE: Information is added in Table 1 legend. With regards to the average degree of regulation, we were not quite sure if this would be the average of all log2foldchange values for individual mt-ncRNAs for each comparison. If it is so, it would provide only less detail. Instead, we have provided the minimum and maximum log2foldchange values for each pairwise comparison in Table 1.

  • Line 167: The feature of overlapping reading frames also constitutes a limitation of the software, because the short reads from the next generation sequencing does not give information about the start and end of the non-coding/alternative RNAs.

RESPONSE: The short reads from NGS are mapped to the mitochondrial genome which gives the information of start position and then using the length of the read , the end position of the ncRNAs are calculated. Then using the annotation information from the GTF file, it is then checked if the ncRNA overlaps a gene or if it is within a gene. So it would be possible to capture all ncRNAs mapping to the mitochondrial genome.

  • Line 186: The software computes alignment to the mtDNA. How many of the hits are also present in nuclear DNA? The short read length makes is likely that the sequences are not unique to the mitochondria. Hence, I believe that the authors must quantify how many of the short RNAs that could also derive from the nuclear genome.

RESPONSE: Yes, most sequences are ambiguous, that is they could be of both nuclear and mitochondrial genomes origin. This is because most animal nuclear genomes, including human genome, contain mitochondrial pseudogenes (NUMTs). In the case of the present datasets: in dataset-1, 563 (27%) mt-sRNAs  map uniquely to the mitochondrial genome and 1552 (73%) map to both nuclear and mitochondrial genomes; in dataset-2, 1293 (14.5%) mt-sRNAs map uniquely and 7587 (85.5%) map to both nuclear and mitochondrial genomes; in dataset-3, 68 (38.2%) mt-lncRNAs map uniquely to the mitochondrial genome and the remaining 110 (61.8%) map to both nuclear and mitochondrial genomes.

However, the functional meaning of NUMTs is negligible. The question whether nuclear genome-encoded sequences contribute to the pool of mitochondrial small RNAs has been investigated and it current understanding that their expression is very low if any at all, and they do not contribute to mt-sRNAs (e.g. Genome Biol Evol. 2019, 11(7): 1883–1896.). It means that the DNA sequence ambiguity in the present study does not considerably affect the expressed gene products, which were investigated. Indeed, in both datsets -1 and -2, only four mitochondrial sRNAs were mapping to genomic meaningful regions (tRNAs), which is a very low percentage, and these sequences were all 15-16 nt in length,  where a probability for random multiple mapping, given the genome size, is relatively high.

We discuss NUMTs in the last paragraph of the Discussion section. Also for the Reviewer’s reference, we attach additional files with the detailed statistics of biotypes and frequencies of nuclear genome-mapped mt-sRNA sequences.

  • Line 317: lncRNAs are normally defined as being above 200nt and having no coding potential. From Additional File 8 it seems like lncRNAs are defined somewhat different (>87nt)

RESPONSE: Yes, indeed. lncRNAs are defined as having length above 200nt. However, certain publications categorize ncRNAs that are 50-200 nt as mid-size or intermediate size RNAs. These ncRNAs can also be identified using lncRNA option in mtR_find. For interested users who also want to study ncRNA that are in the range of 50 to 200 nt, can make use of the default option. Whereas, if users strictly want to study ncRNAs that are greater than 200 nt, they can use the “--filter” option. This parameter is optional. Necessary correction is made in the manuscript (89 to 93).

Round 2

Reviewer 1 Report

The manuscript is much improved, and the authors addressed all the specific critiques we raised. 

Unfortunately, I am still unable to successfully test the tool.  After downloading the updated code from github and one of the samples from the dataset-1 the authors used in the manuscript (Uninfected-3-GGCTAC, SRR1273998.fastq), when I run the tool I get the following error:

$ python3 Test_dir/mtR_find.py hsa sRNA --suspend yes

Traceback (most recent call last):

  File "Test_dir/mtR_find.py", line 738, in

    allmtseq["temp_e"] = allmtseq.astype(str)

  File "/usr/local/lib/python3.10/dist-packages/pandas/core/frame.py", line 3968, in __setitem__

    self._set_item_frame_value(key, value)

  File "/usr/local/lib/python3.10/dist-packages/pandas/core/frame.py", line 4098, in _set_item_frame_value

    raise ValueError("Columns must be same length as key")

ValueError: Columns must be same length as key

I would recommend in the future the authors provide a dockerized version of the tool for ease of deployment and that they test it on multiple computers to ensure portability.

Author Response

Unfortunately, I am still unable to successfully test the tool.  After downloading the updated code from github and one of the samples from the dataset-1 the authors used in the manuscript (Uninfected-3-GGCTAC, SRR1273998.fastq), when I run the tool I get the following error:

$ python3 Test_dir/mtR_find.py hsa sRNA --suspend yes

Traceback (most recent call last):

  File "Test_dir/mtR_find.py", line 738, in

    allmtseq["temp_e"] = allmtseq.astype(str)

  File "/usr/local/lib/python3.10/dist-packages/pandas/core/frame.py", line 3968, in __setitem__

    self._set_item_frame_value(key, value)

  File "/usr/local/lib/python3.10/dist-packages/pandas/core/frame.py", line 4098, in _set_item_frame_value

    raise ValueError("Columns must be same length as key")

ValueError: Columns must be same length as key

RESPONSE: Thank you very much for spotting the error, and we are very sorry it didn’t work. We were able to reproduce the error and fix it. The reason was in incompatibility of  pandas versions (the syntax in line 738 was incompatible with the recent 1.5.3. version of pandas). Now it works with both versions of pandas. A video showing the test run of dataset SRR1273998.fastq with the recent version of pandas (1.5.3) is attached for Reviewer’s reference. While testing with the recent version of pandas, we also identified a small bug in the computation of mt-sRNAs with non-templated CCA additions. Dataset-1 and dataset-2 was re-run using mtR_find tool, there was a slight change in the number of mt-sRNAs with non-templated CCA additions, differentially expressed mtsRNAs, and tRFs not present in Mintbase. Appropriate changes have been made in lines 100-101, 145, and Table 1 of the manuscript. Also, Additional files 6,7,9, 10 and 11 have been updated.

Round 3

Reviewer 1 Report

I was able to run the code now, though I didn't exactly get the same results. My guess is that the tools or versions used in the preprocessing steps are different, though it could be other reasons. Please check. Please document what you used either in the paper or at the tool's website. For a tool paper, it is very important that the users can run the code and can also repeat your experiments as a positive control, before applying on their own data. 

Author Response

Thank you very much for testing the tool and finding discrepancies in the control dataset output. Indeed, it comes as a consequence of pre-processing of a dataset. To maintain the reproducibility of a downstream tool, such as mtR_find , the input adapter-trimmed FASTQ files should be the same, meaning that the raw FASTQ data should be trimmed using the same adapter trimming tool (and its version), and the same parameters. If there is a variation in the parameters or the adapter tool used, this could change the number of unique sequences and the number of reads, which would reflect in the number of mtsRNAs and their corresponding read count. This feature can be checked using the output file “total_count_per_file.csv” from mtR_find, which gives the number of reads in each library.

We used CUTADAPT v1.5 for adapter trimming the test datasets. However, we observed a small difference in the number of adapter-trimmed output reads between cutadapt 1.5 and cutadapt 4.2 on the test datasets. According to Reviewer’s suggestion, adapter-trimmed files, which we used for testing mtR_find, are available for download as compressed files via downloadable link in the GitHub readme page. We have also added a directory named “test” in the tool page for all the three test datasets, which contains the output from mtR_find for the three test datasets (for users to compare the results).  Also the “readme” file under the test directory (https://github.com/asan-nasa/mtR_find/blob/master/test/readme.md) has the information on the tool that we used for adapter trimming and the version of cutadapt tool we used. This information is also appended to the main “readme” section. This can ensure reproducibility of the results using positive control.